

# Ecotoxicological effects of CuO and TiO$_2$ nanoparticles dietary exposure on the marine gastropod *Littorina brevicula*

Sergey Kukla[1], Victor Chelomin[1], Andrey Mazur[1], Nadezhda Dovzhenko[1], Valentina Slobodskova[1] and Evgeniy Elovskiy[2]

[1] V.I. Il'ichev Pacific Oceanological Institute, Far Eastern Branch, Russian Academy of Sciences, Vladivostok, Russia

[2] Far East Geological Institute, Far Eastern Branch, Russian Academy of Sciences, Vladivostok, Russia

## ABSTRACT

Contamination of the aquatic environment by nanoparticles is a threat to marine biota but remains poorly understood. Engineered nanoparticles tend to rapidly sediment in the aquatic environment. Once deposited on the bottom, they become less available to filter organisms, but become available to the bottom feeders and grazers, benthic organisms. In this context, the present study investigated the effects on the gastropod *Littorina brevicula* of a food substrate containing copper oxide and titanium dioxide nanoparticles (NPs) by evaluating metal accumulation in their tissues, cytotoxicity, lipid peroxidation products and genotoxicity. The results showed an increase in copper and titanium content in the soft tissues of *L. brevicula* after 14 days of exposure. Significant cytotoxicity and an increase in malondialdehyde concentration, an indicator of peroxidation of membrane lipid peroxidation, were observed. The results of the comet assay showed pronounced genotoxicity of both NPs, as reflected by an increase in the mean percentage of DNA in the comet tail, as well as an increase in the number of highly damaged comets. The results provided clear evidence that even though the nanoparticles penetrated the digestive system of the mollusk as part of the food substrate, they retained toxic properties. In addition, the food model used in the experiments may be a useful tool in ecotoxicological studies using gastropods and other organisms with similar feeding behavior.

## INTRODUCTION

Since the second half of the last century, nanoparticles have rapidly traveled a long way from a theoretical concept to real applications in very different areas of human life (*Santamaria, 2012*; *Baig, Kammakakam & Falathabe, 2021*). Today, the term ''nanoparticle'' (NP) refers to a particle with linear dimensions of less than 100 nm (*Gambardella & Pinsino, 2022*). Reaching this size, the particles begin to exhibit new catalytic, thermal, magnetic, optical, and other properties not typical for large-sized particles (*El-Kady et al., 2023*). In addition, as the linear size decreases, the active surface area of the particle increases, resulting in NPs being more reactive compared to larger particles of the same material. Due to these

Corresponding author
Andrey Mazur, mazur.aa@poi.dvo.ru

physicochemical properties, NPs are actively used in all spheres of human life, and their production and consumption rates are steadily increasing (*Ahamed et al., 2021*; *El-Kady et al., 2023*). Individual countries invest hundreds of millions of dollars in nanotechnology-related projects today, and the global market for NPs reaches billions of dollars (*Goswami et al., 2017*; *El-Kady et al., 2023*).

Copper oxide and titanium dioxide (CuO and $TiO_2$) NPs are among the most widely used NPs in the world. $TiO_2$ NPs are used in paint products and sunscreens (*Kaegi et al., 2008*; *Gondikas et al., 2014*). CuO is used in electronics, mechanical engineering, *etc.* (*Chang et al., 2012*). Also, their high active surface area and photoactive properties make them effective in removing microorganisms, bacteria, fungi, algae, viruses, and organic pollutants from water (*Chang et al., 2012*; *Amin, Alazba & Manzoor, 2014*). In addition, $TiO_2$, CuO NPs are actively used in the textile industry to impart sun protection and antimicrobial properties to textiles (*Dastjerdi & Montazer, 2010*; *Keller et al., 2013*). However, there is a risk that the increased production and consumption of products containing NPs will lead to their release into the environment. To date, there are many studies showing the release of NPs from widely used products such as cosmetics (*Gondikas et al., 2014*), textiles (*Keller et al., 2013*), paint products (*Adeleye et al., 2016*), packaging materials (*Bumbudsanpharoke & Ko, 2015*), and household appliances (*Farkas et al., 2011*). This has resulted in the current detection of NPs, not only in wastewater treatment plant waters, but also in other aquatic ecosystems (*Farkas et al., 2011*; *Gondikas et al., 2014*; *Zhao et al., 2021*). In addition, previous studies have shown that NPs are contained in the organisms that inhabit them (*Souza et al., 2019*). According to long-term predictions of NP content (such as CuO, Ag, and $TiO_2$), depending on the environment under study, their concentrations can reach values 10 µg/l (in water) and 10 mg/kg (in sediments) (*Zhao et al., 2021*). In general, the penetration of NPs into all parts of ecosystems, including humans, is a major concern among both ecotoxicologists and government authorities (*Santamaria, 2012*).

A significant part of NPs discharged from various materials is thought to get into rivers and terrigenous watercourses and end up in the marine environment. Therefore, in the recent literature, there is a growing number of publications that study various aspects of the interaction of NPs with marine organisms (*Canesi & Corsi, 2016*; *Gambardella & Pinsino, 2022*). Particular attention of ecotoxicologists is focused on littoral zone species, *i.e.,* invertebrates that demonstrate a tendency to accumulate various xenobiotics, including NPs. The major part of studies is carried out on various representatives of bivalves, which are filtrators and show high sensitivity to various pollutants (*Canesi et al., 2010*).

However, despite the growth of publications revealing various aspects of toxic effects of NPs on littoral marine invertebrates and real achievements in this matter, the nature of the influence of NPs in the composition of food substrate remains unclear. It is known that metal oxide NPs in aquatic, especially in marine environments show a tendency to aggregate and agglomerate with suspended organic and inorganic matter, which leads to their sedimentation (*Dwivedi et al., 2015*; *Goswami et al., 2017*; *Abdel-Latif et al., 2020*). In this case, sedimented NP agglomerates become inaccessible to filter feeding organisms, but pose a threat to a wide range of benthic organisms feeding on the deposited organic matter, *i.e.,* bottom feeders and grazers.

Gastropods are one of such organisms that scrape off deposited organic matter. However, studies on the interaction of NPs with gastropods in aquatic environments are rare. As an example, Caixeta and colleagues estimated that between 2010 and 2019, only approximately 60 original articles were published on the toxicity of NPs to gastropods, and only 15% of the papers studied "dietary" impacts (*Caixeta et al., 2020*). Though the "dietary" impact is more universal, as it has been shown that in the case of NP dietary uptake, the chemistry of the surrounding aquatic environment has virtually no effect on their uptake (*Oliver et al., 2014*). Therefore, the issue of biological activity of NPs entering into the digestive system of the organism as a part of food substrate is important but insufficiently studied. It is particularly important to evaluate the biological activity of NPs at the cellular and molecular levels, which are the basis for understanding the mechanisms of development of pathological processes.

Thus, the aim of the present work was to show the effects of NPs dietary exposure on marine gastropods on the example of *Littorina brevicula,* which is a widespread species on the littoral fringe of the temperate coast of the northwestern Pacific (*Azuma & Chiba, 2016*). To understand the issues of their possible toxicity and uptake, our work investigated the high concentration of two types of metal oxide NPs added to a natural organic polymer. The food substrate was agar, one of the components of algae that form the basis of the diet of *L. brevicula*. This food model was successfully applied earlier when studying the effect of microplastic particles on the mollusk *L. brevicula* (*Chelomin et al., 2023*). Cytotoxicity analysis, malondialdehyde content as one of the most frequently used indicators of oxidative stress (*Krishnamurthy et al., 2024*), as well as comet assay, a universal indicator of genotoxic potential of pollutants (*Jiang et al., 2023*), were used as criteria of negative effects.

## MATERIAL AND METHODS

### Analysis of the size of primary NPs

The size of primary particles and their agglomerates was determined by transmission electron microscopy (TEM) according to the method described earlier in *Kukla et al. (2021)*. First, 10 μl of stock solutions of NPs ($TiO_2$ and CuO) were placed on a TEM grid covered with formvar for 30 min, then the filter paper was removed and dried at room temperature. The obtained sample was examined using a Carl Zeiss Libra 200 transmission electron microscope (Carl Zeiss NTS GmbH, Oberkochen, Germany) at 200 kV.

### Description of the experiment

The adult gastropods *Littorina brevicula* (200 individuals) were collected in Alekseev Bay (Peter the Great Bay, Sea of Japan). *L. brevicula* were then placed in the laboratory for further acclimatization (5 d). Acclimatization was performed at a temperature of $17 \pm 0.5\ °C$, with forced aeration and daily water changes. The mollusks were not fed before the experiment, after acclimatization, the mollusks were randomly divided into three groups (control and 2 experimental) with 45 individuals in each. The gastropod of each group was randomly divided into 15 individuals and kept in three parallel round glass tanks with a volume of five l in 2.5 l of seawater. During the experiment, the mollusks were fed a gel plate in the form of a round slide with a diameter of seven cm (volume four ml) attached to the bottom

of the tank. The experiment period was 14 d. Water was changed every 24 h. The aeration and relative temperature of $17 \pm 0.5$ °C were kept constant throughout of the experiment. The gel plates were changed once on the 7th d. During the period of acclimation and the experiment, there was no mortality of *L. brevicula*. The individuals not used in the experiment were released into the natural environment.

On the 14th d, the digestive gland of the mollusks was sampled for further studies. From each tank for each of the three replicates, nine mollusks were taken for determining the level of DNA molecule damage, cytotoxicity assay, and malondialdehyde (MDA) content, six mollusks were used for determining metal content. All procedures in the present work, as well as the mollusks disposal methods, were approved by the Commission on Bioethics at the V.I. Il'ichev Pacific Oceanological Institute, Far Eastern Branch of Russian Academy of Science (protocol No 29 and date of approval 26 June 2024), Vladivostok, Russia.

## Gel plate preparation

The food substrate was a gel plate prepared based on the method proposed by *Odintsov, Karpenko & Karpenko (2023)*. To prepare the plate, aqueous extract of dried seaweed (*Porphyra*) (Sin Young Food Co., Republic of Korea) was mixed with 2% agar-agar (Sigma-Aldrich) in a 1:1 ratio. $TiO_2$ NPs (Sigma-Aldrich) were added to the gel plates of the first experimental group and CuO NPs (Sigma-Aldrich) to the second group at a ratio of 10 mg of NPs per one ml of gel. This concentration was chosen based on several studies investigating the dietary exposure of NPs on gastropods (*Caixeta et al., 2020*). The gel plates of the control group did not contain NPs.

## Determination of Cu and Ti content by inductively coupled plasma mass spectrometry

Dry digestive gland suspensions were soaked in 5 ml of extra-pure concentrated $HNO_3$ acid with the addition of 25 μl of extra-pure concentrated acid. The latter reagent was added for the complete decomposition and stabilization of the final Ti solution. The mixture was then allowed to stand for five days under normal conditions for the primary reactions of degradation of the sample organic matter, and removal of excess $NO_2$ vapor. This procedure was carried out immediately in HP500 autoclaves of the Mars 5 microwave decomposition system (Cem Corporation).

The second stage—tightly closed autoclaves were exposed to microwave radiation. The maximum power of the magnetron was 1200 W, and in the process of decomposition in automatic mode with control "by pressure" was changed by the software of the oven itself. Heating of the reaction mixture up to the set maximum pressure of 160 psi (sensor ESP–1500 Plus) was carried out within 14 min. The maximum temperature was 130 °C (sensor RTP–300 Plus). Further, at an achieved pressure of 158–165 psi, microwave soaking was performed for 14 min. The autoclaves were opened when the temperature was below 40 °C and the pressure was below 58 psi. The decomposed samples were transferred into 50 ml polypropylene tubes using deionized water.

Elemental analysis of biological samples after decomposition was performed by inductively coupled plasma mass spectrometry in conventional (not designed for the

introduction of solutions with high salt content) mode on an Agilent 7,700x spectrometer. Ti was detected in helium low-energy (4.3 ml/min helium flow; three V potential barrier) mode at mass 47, and Cu was detected in mode with a passively operating octopole reaction system (without introduction of any collision and/or reaction gas) at mass 63. The internal standard $^{115}$In was used in these modes. For dilution 500 in terms of the original $3^{\star}\sigma$ sample, the detection limits were 140 and 225 ng/g, respectively.

## Determination of cytotoxicity

The extracted digestive gland was placed in one ml of phosphate-buffered saline (PBS, T = 4 °C, pH = 7.4) and gently cut with surgical scissors. It was then gently mixed by shaking for 15 min to obtain a suspension of individual cells of the digestive gland. The obtained suspension was filtered through a sieve (d = 45 μm) and used for cytotoxic examination. The cytotoxicity of CuO and TiO$_2$ NPs was determined using the resazurin cytotoxicity assay and the neutral red (NR) assay. Cell viability was expressed as a percentage of cell survival compared to control.

The resazurin cytotoxicity assay was carried out according to the method described earlier in *Czekanska (2011)*, with minor modifications. The core principle of this assay rests on the capacity of living cells to transform resazurin into a pink, luminescent product, resorufin. Healthy cells continuously convert resazurin into resorufin, so a change in the total fluorescence of the cell suspension allows us to determine a change in the viability of its cells. For the assay, a 30 μL aliquot of a 1:10 diluted solution of resazurin in phosphate-buffered (PBS) saline at pH 7.4 was added to a total volume of 300 μL of the previously received cell suspension. Following this, the mixture was incubated for 1 h at a temperature of 37 °C on a thermoshaker TS-100C (Biosan, Riga, Latvia). Then, colorimetric analysis was performed at wavelengths of 570 and 600 nm on a UV-2550 spectrophotometer (Shimadzu, Kyoto, Japan).

The NR assay used according to the procedures previously described by *Tang et al. (2022)*, with minor modifications. This method relies on the process of dye uptake and accumulation within lysosomes of healthy cells. In contrast, damaged and dyind cells fail to retain the dye for a long time. For the analysis, 300 μl of cell suspension was prepared, after which 60 μl of TC solution (50 μg/ml) was added there. The mixture was then incubated for 1 h at 37 °C with continuous mixing using a thermoshaker TS-100C (Biosan, Riga, Latvia). After the incubation period, the cells were washed twice with 300 μl of PBS with a pH of 7.4 to remove any remaining dye. To conclude, 300 μl of solution of acetic acid and ethanol (of 1% acetic acid and 50% ethanol) was prepared. The mixture was then incubated at 20 °C for 15 min. The absorbance (at 540 nm) of the mixture was subsequently measured using a UV-2550 spectrophotometer (Shimadzu, Kyoto, Japan).

## MDA content measurement

The MDA content of the digestive gland was determined by color reaction with thiobarbituric acid (TBA, Merck KGaA, Darmstadt, Germany. CAS-No. 504-17-6) based on *Buege & Aust (1978)*. For analysis, the digestive gland was homogenized in 0.1 M phosphate buffer pH 7.0. Measurements were taken at 580 nm and 532 nm wavelength,

then the difference in optical density readings was found. A molar extinction coefficient of 1.56–105/cm/M was used to calculate the MDA content. The relative MDA content was expressed in moles per mg of tissue crude weight. Measurements were performed using a Shimadzu UV-2550 spectrophotometer.

## Comet assay

Damage to DNA was assessed using a modified alkaline version of the DNA comet assay described earlier in *Mitchelmore & Chipman (1998)*. Initially*, L. brevicula* digestive gland cells were isolated from the tissue using calcium- magnesium free buffer (500 mM NaCl, 1,25 mM KCl, 5 mM EDTA-Na$_2$ and 20 mM Tris–HCl, pH 7.4). The resulting cells were immobilized in 1% low-melting agarose (MP Biomedicals, Eschwege, Germany) for subsequent coating of glass slides pre-coated with 1% normal-melting agarose (MP Biomedicals, Eschwege, Germany). To lyse the digestive gland cells, the prepared gel slides were placed in cold high-salt buffer (2.5 M NaCl; 0.1 M EDTA-Na$_2$; 1% Triton X-100; 227; 10% DMSO; 0.02 M Tris–HCl, pH 10) for 1 h. The next step included alkaline treatment (300 mM NaOH, 1 mM EDTA-Na$_2$) of the gel slides for 40 min to detect single-strand DNA breaks and alkali-labile regions. After the alkaline treatment, electrophoresis was performed in the same solution at two V/cm for 20 min. At the end of electrophoresis, the gel slides were neutralized (0.4 M Tris–HCl, pH 7.4) and fixed with ethanol; after air drying, the slides were stained with SYBR Green I.

First, the prepared and colored slides were photographed using an Axio Imager A1 fluorescence microscope (Carl Zeiss, Oberkochen, Germany) equipped with an AxioCam MRc camera (Carl Zeiss, Oberkochen, Germany). The digital photographs were then analyzed using comet analysis software V1.2.2. ASP (Wroclaw, Poland, https://casplab.com, date of circulation March 20, 2025).The parameter "% of DNA in the tail of the comet" was chosen as the most appropriate marker of the degree of DNA damage. In total, 900 comets ($n = 900$) from each group (for one individual = 100 comets) were randomly counted.

## Statistical analysis

The experiment results were processed with MS Excel 2003 and Statistica 10 software package (StatSoft, Tulsa, OK, USA; http://statsoft.ru/resources/support/download.php, accessed on 2 May 2023). For the data, nonparametric Kruskal–Wallis analysis of variance (ANOVA) followed by pairwise Mann–Whitney tests were performed. A difference of $p < 0.05$ was considered statistically significant.

# RESULTS

## NPs characteristics

The TEM results showed that both CuO and TiO$_2$ NPs form large agglomerates in water (Fig. 1). The average size of the agglomerates was 300–500 nm. However, the agglomerates consisted of inividual particles of about 50 nm in size for Cu NPs and 20–30 nm for TiO$_2$ NPs, which corresponds to the size stated by the manufacturer. Also, our data correlate with the results obtained by the authors working with the fully identical commercial particles (*Hu et al., 2014*; *Ali & Ali, 2015*) (Table 1).

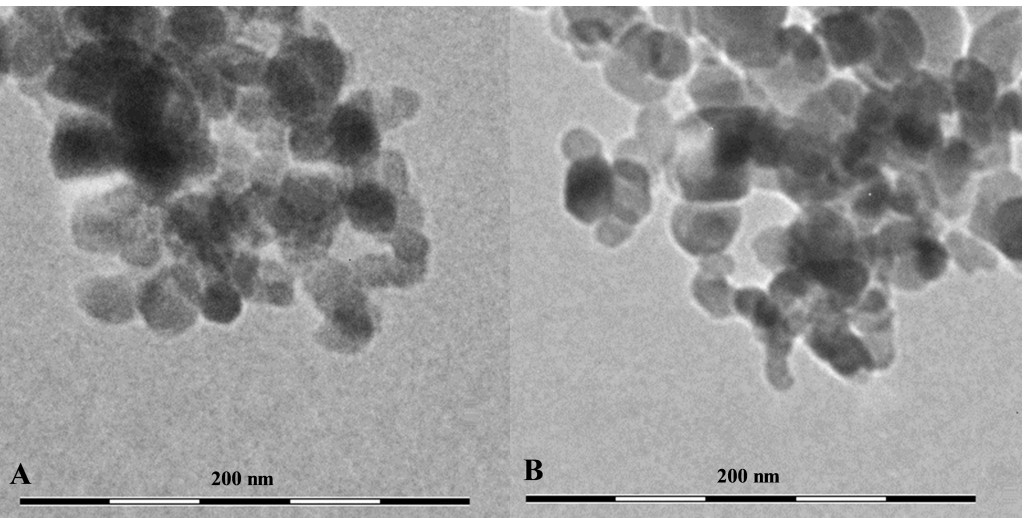

**Figure 1** **The shape and size of using NPs.** (A) CuO NPs, (B) TiO$_2$ NPs.

**Table 1** **The physicochemical properties of using NPs.**

| NP | Purity, % | Particle size, nm | Hydrodynamic radius, nm | BET, m$^2$/g | Zetta potential, mV |
|---|---|---|---|---|---|
| CuO | 100 | 50 | 100 | 29 | −57 |
| TiO$_2$ | 99.5 | 21–34 | 190.5 | 35–65 | −13.9 |

## Inductively coupled plasma mass spectrometry

Mollusks of control and experimental groups ate agar plates with algal extract, and the presence of both types of NPs (CuO and TiO$_2$) in the food substrate resulted in the accumulation of the corresponding elements in the soft tissues of *L. brevicula*. Within two weeks, in mollusks whose food was added CuO NPs, the copper concentration in soft tissues and in the digestive gland increased more than 5-fold, reaching values of 320–371 µg/g dry weight. In similar experiments with the addition of TiO$_2$ NPs, the titanium content in the digestive gland of experimental mollusks increased more than 50 times, and in soft tissues—more than 100 times, compared to the control group, and amounted to 318 and 468 µg/g dry weight, respectively (Fig. 2).

## Cytotoxicity

The accumulation of both tested NPs in the digestive gland of *L. brevicula* induced a cytotoxic reaction in its cells (Fig. 3). The resazurin assay showed that, compared to control mollusks, specimens exposed to CuO and TiO$_2$ NPs significantly decreased the ability to recover resazurin into resorufin. These results indicate a decrease in the metabolic activity of digestive gland cells from both experimental groups of *L. brevicula*. Different trend was observed when the NR test was used. The results show that digestive gland cells only from the mollusks exposed to TiO$_2$ NPs have a significantly decreased ability to retain the dye, indicating destabilization of lysosome membranes induced by uptake of TiO$_2$ NPs.

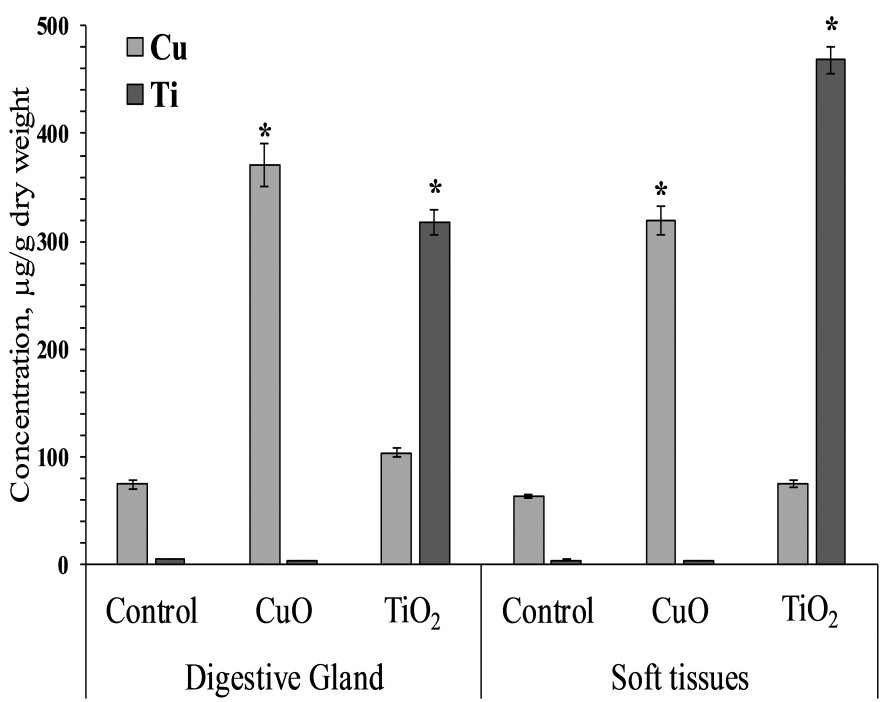

**Figure 2** **Cu and Ti concentration in digestive gland and soft tissues of experimental and control group of *L. brevicula* (mean ± standard deviation).** * Difference from the control is significant ($p < 0.05$).

In general, results of cytotoxicity tests shown in both experimental groups of mollusks indicate the suppression of general metabolism in these cells.

## MDA level

The experimental results showed that MDA content in the digestive gland cells of the mollusks control group was $47.08 \pm 2.32$ nmole/g wet weight, whereas in the experimental *L. brevicula* whose food was added with CuO and $TiO_2$ NPs, this index increased by 25% and was $60.26 \pm 3.73$ and $60.58 \pm 2.49$ nmole/g wet weight, respectively (Fig. 4).

## DNA damage

The level of nuclear DNA damage in the cells of the control group of mollusks was $4.19 \pm 0.73\%$, but after feeding with food containing CuO NPs, this parameter increased almost twofold compared to the control group, and was $8.61 \pm 1.19\%$. *L. brevicula* fed with $TiO_2$ NPs, there was also a significant increase in the level of DNA damage, which amounted to $6.39 \pm 0.85\%$ (Fig. 5A). The destabilising tendency is more evident when the whole array of formed comets is distributed into groups according to the level of DNA fragmentation (with an interval of 3%) (Fig. 5B). From the analysis of Fig. 4B, it follows that *L. brevicula* is characterised by high genome stability, as evidenced by the large proportion of cells (about 70%) with a very low level of DNA fragmentation (less than 3%) and the absence of cells in which DNA destruction was more than 20%. In experimental mollusks

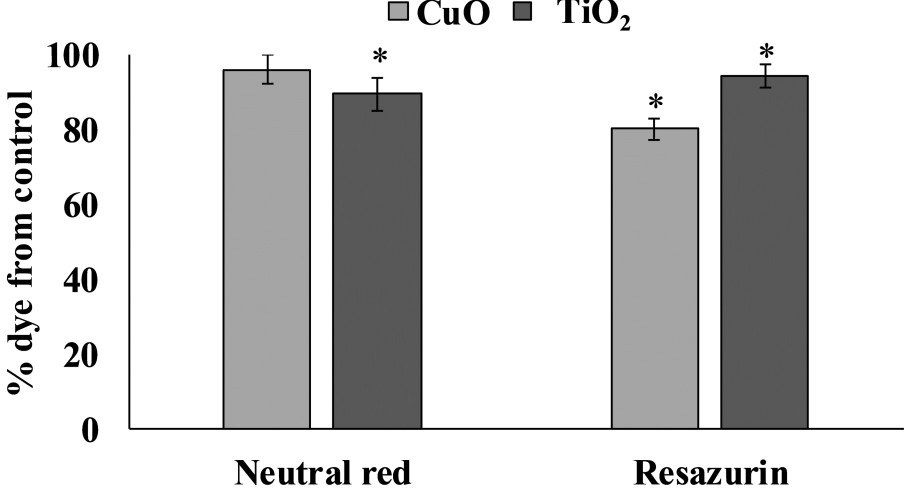

**Figure 3** Determination of viability of *L. brevicula* digestive gland cells in the control and experimental groups using a resazurin test and neutral red dye (mean ± standard deviation). *—difference from control is significant at $p < 0.05$.

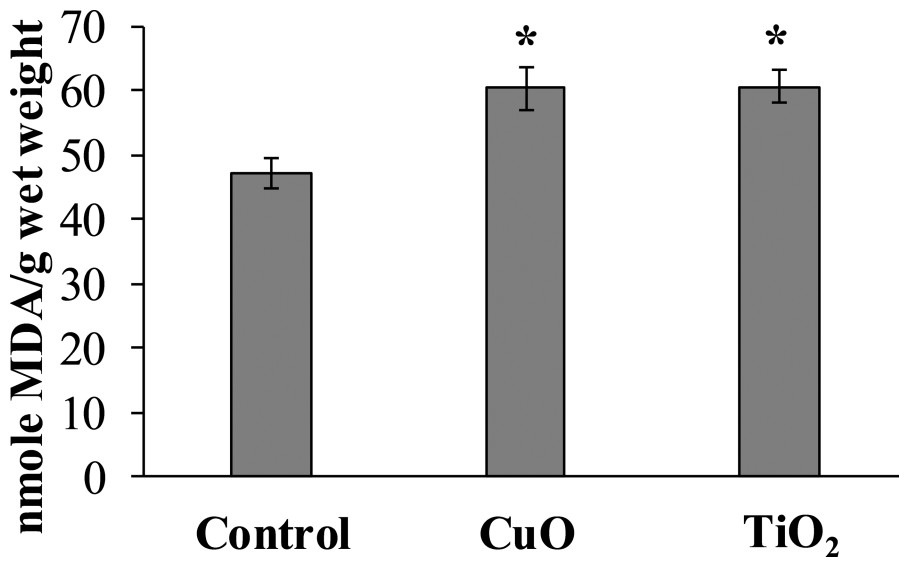

**Figure 4** Effect of CuO and TiO$_2$ NPs on the levels of MDA in cells of the digestive gland of *L. brevicula* (mean ± standard deviation). *—difference from control was significant at $p < 0.05$.

after feeding with CuO and TiO$_2$ NPs, not only the proportion of cells with a low level of DNA damage decreased significantly to 33% and 45.7%, respectively, but also comets (15.5% and 9.3%, respectively) with DNA destruction levels in the range of 20 to 40% were recorded.

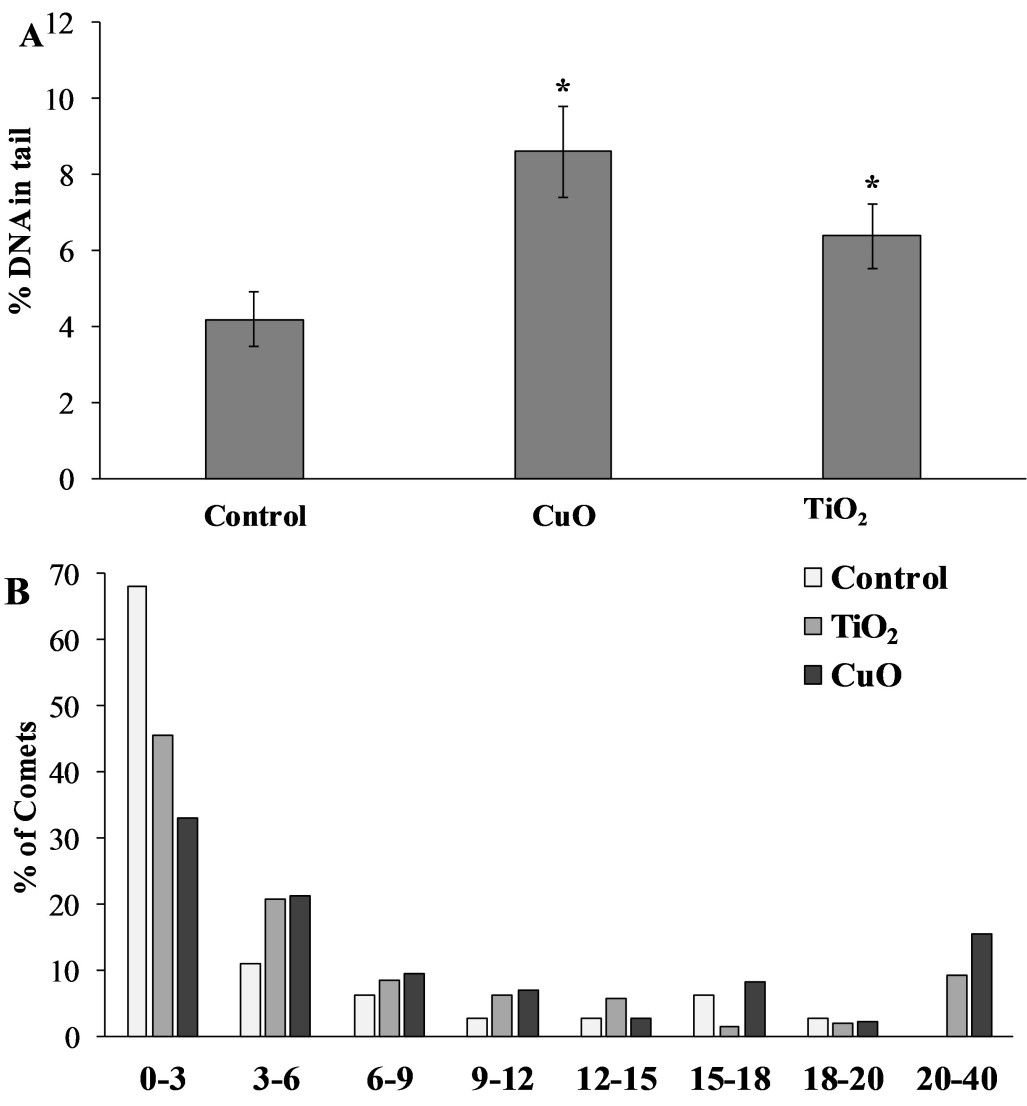

**Figure 5** (A) Level of nuclear DNA damage in *L. brevicula* digestive cells after feeding food containing CuO and TiO$_2$ NPs (mean ± standard deviation) (B) Distribution of comets according to the degree of nuclear DNA damage. *—difference from control was significant at $p < 0.05$.

## DISCUSSION

The results of analyzing the dimensional characteristics of the NPs suspension before its introduction into the agarose gel showed a high degree of they aggregation. The fact that NPs are susceptible to aggregation when released into the aquatic environment is well known in the scientific literature. A combination of various biotic and abiotic factors of the water environment can both cause and limit NPs, intensive aggregation (*Ameh & Sayes, 2014*; *Zhu et al., 2014*). However, the electron microscope images taken as part of this study indicated that, despite the size of the aggregates being several times greater than 100 nm, they were composed of primary particles whose dimensions were in the "nanoscale" range.
*Littorina brevicula* is a typical herbivorous representative of gastropod mollusks, widely distributed among benthic communities of the littoral zone of the Sea of Japan. Due to its specific mouth structure, *L. brevicula* scrapes off all adhering particles together with the phytosubstrate. Therefore, the method of feeding *L. brevicula* with NPs containing food substrate described in our work essentially simulates the natural way of feeding of these mollusks, and provides an appropriate model for characterising the bioavailability of these metal oxides forms.

As shown by mass spectrometry results, *L. brevicula* appeared to be able to uptake NPs from the NPs containing substrate. As a result, an increase in the content of titanium and copper was observed both in the digestive gland of the mollusk and in its other soft tissues. Application of this food model indicates the potential of gastropod mollusks with similar feeding patterns to uptake and accumulate high amounts of metal oxide NPs, which presents an ecological risk in terms of NP transport through the trophic chain.

In this regard, our results are in good agreement with the results of previous studies in which representatives of marine and freshwater gastropod mollusks actively uptake and accumulate various inorganic NPs, including CuO and $TiO_2$, from the environment (*Li, Turner & Brown, 2013*; *Pang et al., 2013*; *Ramskov et al., 2015*; *Ma, Gong & Tian, 2017*; *Silva et al., 2022*; *Wang et al., 2022*).

For example, *Li, Turner & Brown (2013)* showed that periwhinkle *L. littorea* absorbed Ag NPs both from the aquatic environment and from the alga *Ulva lactuca*, which was pre-incubated in water with nanoparticles. The presence of Ag NPs in sediment also led to an increase in Ag concentration in the gonads of the freshwater snail *Bellamya aeruginosa* (*Bao et al., 2018*). The freshwater snails *Physa acuta*, *Lymnaea stagnalis* and *Planorbarius corneus* also actively absorbed Ag NPs and accumulated Ag in soft tissues (*Silva et al., 2022*; *Wang et al., 2022*). Also, species specificity in the rate of metal accumulation was observed, apparently related to different physiological activity of mollusks (*Silva et al., 2022*).

CuO NPs mixed with sediments were available for freshwater mollusks (*Bellamya aeruginosa*, *Potamopyrgus antipodarum*) and concentrated in the digestive gland, gonads, and muscle tissue (*Pang et al., 2013*; *Ramskov et al., 2015*; *Ma, Gong & Tian, 2017*). At the same time, Ramskov et al. noted that copper accumulated in tissues was not excreted even after 14 days of keeping snails in a medium containing no CuO NPs (*Ramskov et al., 2015*). In a similar series of experiments, titanium accumulation in the tissues of snails *Cipangopaludina chinensis* and *B. aeruginosa* was also observed when $TiO_2$ NPs was added to the medium (*Yeo & Nam, 2013*; *Ma, Gong & Tian, 2017*).

Furthermore, several studies have reported that after entry into the body, NPs are able to cross tissue barriers (*Brun et al., 2014*; *Sawicki et al., 2019*; *Bongaerts et al., 2020*), including penetrating into the haemolymph of gastropod mollusks (*Bobori et al., 2020*). However, studies involving different types of inorganic NPs show that in gastropod mollusks, the digestive gland is the main site of concentration of the metal oxides nanoforms (*Ma, Gong & Tian, 2017*; *De Vasconcelos Lima et al., 2019*; *Kuehr, Kosfeld & Schlechtriem, 2021*; *Wang et al., 2022*). The exact mechanisms for this are unclear, but it has been suggested that it is a result of active endocytosis mechanisms and high levels of metal-binding proteins in molluskan digestive gland cells (*Khan et al., 2015*; *Kuehr, Kosfeld & Schlechtriem, 2021*).

For the ecotoxicological assessment of the NPs environmental impact, the question of the possible consequences of NP accumulation on sensitive biochemical parameters, such as DNA integrity and oxidative stress is important. In ecotoxicological studies, these results can be further used to identify the minimum effective concentrations of a toxicant and compare them with the real concentrations in the environment.

The accumulation of both tested NPs in the digestive gland of *L. brevicula* induced a cytotoxic reaction in its cells. The resazurin assay showed that, compared to control mollusks, specimens exposed to CuO and TiO$_2$ NPs significantly decreased the ability to recover resazurin into resorufin. These results indicate a decrease in the metabolic activity of digestive gland cells from both experimental groups of *L. brevicula*. The intensity of conversion of resazurin to resorufin directly depends on the activity of oxireductases in a living cell. Thus, the results of the resazurin test showed a decrease in the efficiency of energy metabolism inside the cell. Decrease of dye reduction parameters in the NR test in the group feeding on the gel containing TiO$_2$ NPs also indicates a decrease in the viability of mollusks in this group. Similar changes in NR dye retention were observed in the hemocytes of the gastropod mollusks after FeO and TiO$_2$ NPs exposure (*Sidiropoulou et al., 2018*; *Bobori et al., 2020*).

The observed decreasing of viability of *L. brevicula* digestive gland cells was followed by changes suggestive of the development of oxidative stress in it. This is evidenced by the accumulation of the end product of lipid peroxidation—malondialdehyde (MDA), which is an early and sensitive indicator of oxidative stress development (*Prokic et al., 2019*). Although the specific mechanism of oxidative stress induced by NPs is still unclear, there is a generally accepted view that this process is associated with enhanced generation of reactive oxygen species (ROS) (*Ma, Gong & Tian, 2017*; *Dovzhenko et al., 2022*; *Abdel-Azeem, Osman & Mohamed, 2023*). The literature lacks data on the biochemical response of marine gastropods to NP exposure (*Caixeta et al., 2020*). Nevertheless, our results can be compared with data obtained on freshwater and terrestrial gastropods. Thus, in land snails *Monacha cartusiana*, pronounced oxidative stress was observed in the digestive gland as a result of exposure to ZnO NPs according to the results of a battery of biomarkers analyzed (*Abdel-Halim, Osman & Abdou, 2020*). *Bao et al. (2018)* in their work on exposure to Ag NPs mixed with sludge showed that despite low rates of silver accumulation in mollusks, there were significant changes in the levels of catalase, superoxide dismutase and peroxidase, indicating the development of oxidative stress. Copper accumulation, after exposure to sediment containing NPs was accompanied by an increase in MDA concentration in *B. aeruginosa* tissues (*Ma, Gong & Tian, 2017*). In studies on the exposure of the land snail *Helix aspersa* to a food mixture containing FeO NPs for 20 days, an increase in ROS production and protein carbonyl damage in snail hemocytes was noted as early as the first day of exposure (*Sidiropoulou et al., 2018*). Exposure of the terrestrial snail *H. aspersa* to TiO$_2$ NPs resulted in disturbances of antioxidant system in the digestive gland (*Abdel-Azeem, Osman & Mohamed, 2023*). Moreover, besides being a biomarker of oxidative stress, MDA itself is a highly toxic intermediate product of membrane lipid oxidation and its accumulation in digestive cells can results in further toxic effects (*Krishnamurthy et al., 2024*).

In addition to the induction of accumulation of highly toxic MDA, feeding *L. brevicula* with food containing CuO and TiO$_2$ NPs resulted in increased nuclear DNA damage in digestive gland cells, as measured by the comet assay. These results indicate that the absorption and concentration of both NPs affect the biochemical processes that maintain the stability of the digestive gland cell. The results obtained are in agreement with the few studies on DNA damage induced by exposure to inorganic NPs in gastropods. Thus *Ali et al. (2012)*, *Ali (2014)* and *Ali & Ali (2015)* showed dose and time dependent DNA damage in *Lymnaea luteola* after exposure to ZnO, TiO$_2$ and CuO NPs using comet assay. The authors suggest that the cause of DNA damage was oxidative stress, was evidenced by an increase in the level of damage to membrane lipids and a decrease in the activity of antioxidant system enzymes. Similarly, after exposure of the land snail *Cornu aspersum* to feed containing CuO and ZnO NPs, DNA damage and an increase in oxidative stress were detected in the mollusk hemocytes (*Feidantsis et al., 2020*). In addition, exposure to FeO NPs results in an increase in DNA damage in gastropods *H. aspersa* and *C. aspersum* (*Sidiropoulou et al., 2018*; *Kaloyianni et al., 2020*). Using the micronucleus test, *Radwan et al. (2019)* showed that the resulting DNA damage, after NPs exposure on gastropods, remained significant even after a week of complete cessation of exposure. Thus, the DNA integrity damage we have identified poses a serious risk to a population exposed to NP exposure.

Regardless of the specific mechanism, it must be highlighted, that biochemical disorders initiated by NPs are more diverse and are not limited to the membranes lipid peroxidation and genotoxicity identified in our study. Nevertheless, it is necessary to focus on DNA damage, due to the diagnostic and prognostic role of this marker. It is possible that damaged DNA, initiating defects in the biosynthesis of biomolecules, may initiate a cascade of biochemical abnormalities, ultimately leading to serious pathological effects in the organism, especially in coastal and estuarine zones.

## CONCLUSION

In this work, to evaluate the bioavailability and biological activity of metal oxide nanoparticles, we used a food model that most adequately corresponds to the feeding method of gastropod mollusks. It was shown that the introduction of CuO and TiO$_2$ nanoparticles into the food substrate (agar) leads to the accumulation of the corresponding elements in the digestive system of the gastropod *L. brevicula*, especially in the cells of the digestive gland. Despite the fact that the nanoparticles penetrated into the digestive system of the mollusk as part of the food substrate, they retained toxic properties, inducing an increase in the processes of lipid peroxidation and DNA damage in the cells of the digestive gland. The food model used in the experiments can be a useful tool in future ecotoxicological studies using gastropod mollusks and other organisms with a similar feeding pattern.

### Funding

This study was supported by the state assignment for research work of V.I.l'ichev Pacific Oceanological Institute, FEB RAS (No. 124022100077-0). The funders had no role in study design, data collection and analysis, decision to publish, or preparation of the manuscript.

### Grant Disclosures

The following grant information was disclosed by the authors:
V.I.l'ichev Pacific Oceanological Institute, FEB RAS: No. 124022100077-0.

### Competing Interests

The authors declare there are no competing interests.

### Author Contributions

- Sergey Kukla conceived and designed the experiments, performed the experiments, authored or reviewed drafts of the article, and approved the final draft.
- Victor Chelomin conceived and designed the experiments, authored or reviewed drafts of the article, and approved the final draft.
- Andrey Mazur performed the experiments, analyzed the data, prepared figures and/or tables, and approved the final draft.
- Nadezhda Dovzhenko performed the experiments, analyzed the data, prepared figures and/or tables, and approved the final draft.
- Valentina Slobodskova conceived and designed the experiments, performed the experiments, prepared figures and/or tables, and approved the final draft.
- Evgeniy Elovskiy analyzed the data, prepared figures and/or tables, and approved the final draft.

### Ethics

The following information was supplied relating to ethical approvals (*i.e.*, approving body and any reference numbers):

All procedures were approved by the Commission on Bioethics at the V.I. Il'ichev Pacific Oceanological Institute, Far Eastern Branch of Russian Academy of Science.

### Data Availability

The unprocessed measurements are available in the Supplemental File.

### Supplemental Information

Supplemental information for this article can be found online at http://dx.doi.org/10.7717/peerj.19838#supplemental-information.

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
