# Peer review of "Ecotoxicological effects of CuO and TiO2 nanoparticles dietary exposure on the marine gastropod Littorina brevicula"

_PeerJ, doi:10.7717/peerj.19838_

## Round 0.1 · original submission · Minor Revisions

Dear Dr. Mazur,

You can find the comments and suggestions of the expert reviewers in the attached reports. As you will see, expert reviewers have pointed out the errors. Consequently, a minor revision is needed for your article.

I request that you improve your manuscript following the reviewers' suggestions

Sincerely

·

Basic reporting

In their study, the authors investigated the effects of dietary exposure to nanoparticles (copper oxide and titanium dioxide) using cytotoxicity tests, MDA content measurement, DNA damage, comet assay methods for the experimental species Littorine brevicula. Aquatic ecosystems are dynamic structures and are constantly under pressure from different pollutants. The paper is well written, has current and important data. The Introduction and others sections provide useful information for the readers. The paper has a potential to be accepted, but some important points have to be clarified or fixed before we can proceed and a positive action can be taken.
I here summarize this points:
1- The species name in keywords should be written as Littorina brevicula, not L. brevicula.
2- The reasons for choosing Littorina brevicula as the experimental organism should be given in the introduction of the text.
3- How many days of acclimatization period was applied should be given in the materials and methods section (in the text only the information that the molluscs were not fed for 5 days is given).
4- How many individuals were included in the acclimatization period (initial number of individuals). Did any individuals die at the end of the period? This should be explained.
5- If acid was used during digestion of the samples to determine Cu and Ti concentrations, it must be given in the text.
6- If the recovery percentage was calculated using certified reference material or spiked sample in the method validation of elemental analysis, it must be given in the text.
7- Figures 2-5 are prepared with bar graphs. By presenting the data using different graphs, the text can be made visually rich.
8- The reference list must be checked again.

The content of the article in accordance with the aims of the PeerJ
The article is scientifically sufficient.
Keywords are well chosen so that the article can be found by indexes.
The literature has been adequately critical, current and internationally evaluated by the authors.
The language of the article is correct and clear.
The discussion part is not comprehensive in the paper.

Experimental design

The methods applied in the study are adequate and valid.

Validity of the findings

The data were rendered meaningful through statistical analysis.

Additional comments

The experiment was well designed. The data is new. The discussion is very comprehensive.

·

Basic reporting

The article is scientifically substantiated and suitable for inclusion in academic literature. It has a logical structure, a sufficient number of figures and tables. The original data are provided.

Experimental design

Overall, the work is interesting and informative. The topic considered by the authors is relevant, the methods used are up-to-date. In my opinion, after correcting some of the comments below, it may be of interest to a fairly wide range of readers.

Comments on Section 1. Introduction
The authors provide data from 2010-2019 on lines 103-106. However, by 2025, these data are no longer relevant, so it is not correct to refer to them when describing the relevance of the study.

Comments on Section 2. Materials and Methods
In this section, out of 9 references provided, 4 are the authors' works. This amount of self-citation seems excessive. It is better to provide a full description of the methods in the text or put them in Supplemental files.
In paragraph 2.5. Determination of Cytotoxicity (lines 186-188), the authors refer to their own article when describing the methodology. At the same time, the specified methodology is not the author's (the submitted article provides a link to the protocol "Knapp J.L.A., González-Pinzón R., Haggerty R. The resazurin-resorufin system: Insights from a decade of “smart” tracer development for hydrologic applications. Water Res. 2018;54:6877–6889. doi: 10.1029/2018WR023103"). I believe it is correct to refer to the original source of information in the presented article.
The justification for the selected single concentration of nanoparticles is given unconvincingly. A link is given to a review article that presents works with a wide range of concentrations used.

Comments on Section 3. Results
The characterization of the tested nanoparticles was ambiguous. In Section 3.1. the sizes are 20-40 nm, and in Table 1 for CuO - 50 nm, and for TiO2 - 21-34.
In addition, the purity of TiO2 is 95.5%, but nowhere is it indicated what is included in the composition of impurities. It is quite possible that the toxic effect may be associated with the composition of impurities.

Comments on Section 4 Discussions
In the stocks 291-298, the authors write that nanoparticle aggregates consist of individual unbound particles. Based on the definition that the aggregation (agglomeration) of nanoparticles is a set of particles firmly held together, I consider this statement to be erroneous. In addition, aggregation changes the properties of nanoparticles, since the features of aggregation affect the structural organization of the dispersed phase, the packing of particles, their interaction with each other and with the dispersion medium.

Validity of the findings

All underlying data have been provided; they are robust, statistically sound, & controlled - Yes
Conclusions are well stated, linked to original research question & limited to supporting results - Yes

Additional comments

Technical notes
1. Line 134 - °C is written in a different font
2. Line 164 - drop the number in NO2 in the subscript
3. Lines 195, 201, 204 - the ° designation is written in a smaller font and is positioned strangely.
4. Line 273 - there should be a period in the value 47.08.
5. Line 406 - drop the number in TiO2 in the subscript
6. The section numbering is broken (discussions and conclusion are numbered with the number 4)
7. In Fig. 5B, the columns of different nanoparticles are practically indistinguishable by color. And in the legend, the number 2 in TiO2 should be made a subscript.

---

## Round 0.2 · accepted · Accept

Dear Dr. Mazur,
I thank you for making the corrections and changes requested by the reviewers. I read and checked your valuable article carefully and am happy to inform you that the article has been accepted for publication in PeerJ.
Sincerely yours,